

# CY-Bench: A comprehensive benchmark dataset for sub-national crop yield forecasting

Dilli Paudel[1], Michiel Kallenberg[1], Stella Ofori-Ampofo[2], Hilmy Baja[1], Ron van Bree[1], Aike Potze[1], Pratishtha Poudel[3], Abdelrahman Saleh[4], Weston Anderson[5], Malte von Bloh[2], Andres Castellano[6], Oumnia Ennaji[7], Raed Hamed[8], Rahel Laudien[9], Donghoon Lee[10], Inti Luna[11], Michele Meroni[12, 20], Janet Mumo Mutuku[13], Siyabusa Mkuhlani[14], Jonathan Richetti[15], Alex C. Ruane[6], Ritvik Sahajpal[5], Guanyuan Shai[5], Vasileios Sitokonstantinou[11], Rogério de Souza Nóia Júnior[16], Amit Kumar Srivastava[17], Robert Strong[18], Lily-belle Sweet[19], Petar Vojnović[20], and Ioannis N. Athanasiadis[1]

[1]Wageningen University and Research, Artificial Intelligence, PO Box 16, Wageningen, 6700 AA, the Netherlands.
[2]Technical University of Munich, Chair of Data Science in Earth Observation, Arcisstraße 21, Munich, 80333, Germany
[3]Purdue University, Department of Agronomy, 915 Mitch Daniels Blvd, West Lafayette, IN 47907, United States
[4]Ankara University, Faculty Of Agriculture Engineering, Dögol Caddesi 06100 Tandoğan, Ankara, 6110, Turkey
[5]University of Maryland, Department of Geographical Sciences, 7251 Preinkert Drive, Collega Park, MD 20742, United States
[6]NASA Goddard Institute for Space Studies, GISS Climate Impacts Group, Mail Code 611, New York, NY 10025, United States
[7]Mohammed VI Polytechnic University, College of Computing, Lot 660, Benguerir, 43150, Morocco
[8]Vrije Universiteit Amsterdam, Institute for Environmental Studies, De Boelelaan 1105, Amsterdam, 1081 HV, the Netherlands
[9]Potsdam Institute for Climate Impact Research, Department of Climate Resilience, PO Box 60 12 03, Potsdam, 4412, Germany
[10]University of Manitoba, Department of Civil Engineering, 15 Gillson Street, Winnipeg, MB R3T 5V6, Canada
[11]Universitat de València, Image Processing Laboratory, C/ Catedràtic Agustín Escardino Benlloch, 9, València, 46980, Spain
[12]Seidor Consulting, C/Provençals 44, Barcelona, 08019, Spain
[13]International Crops Research Institute for the Semi-Arid Tropics, West and Central Africa Region Hub, PO Box 320, Bamako, Mali
[14]International Institute of Tropical Agriculture, Natural Resources Management, PO Box 30677, Nairobi, 00100, Kenya
[15]Commonwealth Scientific and Industrial Research Organisation (CSIRO), Agriculture and Food, 147 Underwood Av, Perth, WA 6014, Australia
[16]National Research Institute for Agriculture, Food and Environment (INRAE), UMR LEPSE, 2 Pl. Pierre Viala, Montpellier, 34000, France
[17]Leibniz Centre for Agricultural Landscape Research, Simulation and Data Science, Eberswalder Straße 84, Müncheberg, 15374, Germany
[18]Texas A&M University, Agriculture Education, 600 John Kimbrough Blvd, College Station, TX 77843-2116, United States
[19]Helmholtz Centre for Environmental Research, Department of Computational Hydrosystems, Permoserstraße 15, Leipzig, 04318, Germany
[20]European Commission's Joint Research Centre, Food Security Unit, Via E. Fermi 2749, Ispra, VA I-21027, Italy

**Correspondence:** Michiel Kallenberg (michiel.kallenberg@wur.nl)

**Abstract.** In-season, pre-harvest crop yield forecasts are essential for enhancing transparency in commodity markets and improving food security. They play a key role in increasing resilience to climate change and extreme events and thus contribute to the United Nations' Sustainable Development Goal 2 of zero hunger. Pre-harvest crop yield forecasting is a complex task, as several interacting factors contribute to yield formation, including in-season weather variability, extreme events, long-term climate change, soil, pests, diseases and farm management decisions. Several modeling approaches have been employed to





capture complex interactions among such predictors and crop yields. Prior research for in-season, pre-harvest crop yield forecasting has primarily been case-study based, which makes it difficult to compare modeling approaches and measure progress systematically. To address this gap, we introduce CY-Bench (Crop Yield Benchmark), a comprehensive dataset and benchmark to forecast maize and wheat yields at a global scale. CY-Bench was conceptualized and developed within the Machine Learning team of the Agricultural Model Intercomparison and Improvement Project (AgML) in collaboration with agronomists, climate scientists, and machine learning researchers. It features publicly available sub-national yield statistics and relevant predictors—such as weather data, soil characteristics, and remote sensing indicators—that have been pre-processed, standardized, and harmonized across spatio-temporal scales. With CY-Bench, we aim to: (i) establish a standardized framework for developing and evaluating data-driven models across diverse farming systems in more than 25 countries across six continents; (ii) enable robust and reproducible model comparisons that address real-world operational challenges; (iii) provide an openly accessible dataset to the earth system science and machine learning communities, facilitating research on time series forecasting, domain adaptation, and online learning. The dataset (https://doi.org/10.5281/zenodo.11502142, (Paudel et al., 2025a)) and accompanying code (https://github.com/WUR-AI/AgML-CY-Bench, (Paudel et al., 2025b))) are openly available to support the continuous development of advanced data driven models for crop yield forecasting to enhance decision-making on food security.

## 1 Introduction

The global food system faces significant challenges, including unequal access to resources and volatile markets, despite advancements in agricultural production (Ambikapathi et al., 2022; Zhang et al., 2022; Chen and Villoria, 2022; Zelingher and Makowski, 2023; Schneider et al., 2023a) To enhance food security policies, experts have emphasized the need for improved data, maps, and predictions (Mehrabi et al., 2022; Ennaji et al., 2023; Fanzo, 2024). Pre-harvest yield forecasts, in particular, play a critical role in enhancing global market transparency and enabling decision-makers to plan and respond effectively to potential food shortages, especially in the face of a changing climate (Becker-Reshef et al., 2020; Tanaka et al., 2023; Stuart et al., 2024).

Crop yield forecasts are produced by both private entities and government institutes using field surveys, process-based crop models, statistical regression and machine learning (Basso and Liu, 2019; Schauberger et al., 2020; Paudel et al., 2021; Gavasso-Rita et al., 2023). Commonly used predictors are weather, soil moisture, crop productivity, and remotely-sensed vegetation health indicators. Data availability determines the yield forecasting modeling setup and the selected spatial scale, which can range from national to sub-national and field levels. For example, the European Commission's Joint Research Centre (EC-JRC) regularly





produces national crop yield forecasts for the EU and surrounding countries using crop models, agro-meteorological analyses and the expertise of analysts (van der Velde and Nisini, 2019). Data for sub-national crop yield forecasting, which focuses on higher resolution administrative units (e.g., regions, provinces) and captures spatial yield variability within a country (Meroni

et al., 2021; Paudel et al., 2022), is crucial for targeted food security planning. Such data is usually publicly available, but compiling them for many countries is challenging due to differences in collection and reporting protocols, including language and data format.

Traditionally, crop yield prediction has been based on biophysical process based crop models, grounded in decades of agricultural knowledge. Their challenges in parametrization (He et al., 2017; Wallach et al., 2021; Seidel et al., 2018) however

limits scalability and their reliance on first principles hinders significant improvements in forecasting accuracy (van der Velde and Nisini, 2019). Machine learning methods offer promising alternatives, capturing processes not fully covered by biophysical models. However, they typically require high-quality datasets covering large areas and multiple years. Several review articles (Chlingaryan et al., 2018; Kamilaris and Prenafeta-Boldú, 2018; Liakos et al., 2018; Van Klompenburg et al., 2020; Benos et al., 2021; Oikonomidis et al., 2022) have highlighted promising performance of machine learning methods, including deep learning,

for pre-harvest yield forecasting (Schlenker and Roberts, 2009; You et al., 2017; Khaki et al., 2020; Mateo-Sanchis et al., 2021; Paudel et al., 2022; Fan et al., 2022; Liu et al., 2022; Lesk et al., 2022; Paudel et al., 2023b; Vijverberg et al., 2023; Ma et al., 2023; Priyatikanto et al., 2023; Ennaji et al., 2024). However, the data and code used in such studies are not always available, and the diversity in evaluation procedures, metrics, and datasets makes intercomparison and synthesis of results difficult. As a result, the research community is unable to reproduce results and compare the strengths and weaknesses of different methods

across crops and regions.

To better understand the specific strengths and weaknesses of data-driven methods for pre-harvest yield forecasting, and to drive future research progress, well-documented benchmark datasets compiled by domain experts are vital (Tsaftaris and Scharr, 2019; Dueben et al., 2022; Rolnick et al., 2024; Sweet et al., Under review). Benchmark datasets must cover a wide variety of regions and countries (Richards et al., 2023) and reflect the needs of the worldwide community (Tzachor et al., 2022; Nakalembe

and Kerner, 2023). In addition to producing accurate forecasts, models must be reliable in real-world settings for adoption by stakeholders (van der Velde and Nisini, 2019). The evaluation metrics should closely represent the needs of stakeholders and allow a more granular breakdown of model performance (Thomas and Uminsky, 2022; Burnell et al., 2023) - for example, the model's ability to capture yield variability in years with climate extremes (Watson, 2022). To avoid overestimation of model skill, the evaluation procedure must take into account the specific challenges arising from the use of spatio-temporal data that

does not satisfy independent and identically distributed assumptions (Meyer and Pebesma, 2022; Sweet et al., 2023; Kapoor and Narayanan, 2023; Richetti et al., 2023).

A few research works have compiled benchmark-like datasets that include components related to crop yield prediction. SustainBench (Yeh et al., 2021) includes a benchmark dataset for crop yield prediction, and targets end-of-season prediction for only one crop (soybean) in three countries (United States, Brazil and Argentina). Another public dataset is CropNet (Lin

et al.), which only covers the United States. Similarly, there are ongoing efforts to produce a multi-task benchmark dataset which includes yield prediction in the USA as a sub-task (Höhl et al., 2023). Apart from these, other available data contributions



include yield statistics only (Lee et al., Under review; Potter, 2019; Ronchetti et al., 2024; Duden et al., 2024; Argentina; Australia; Brazil; China; India; Mexico) or sample data published with articles (Khaki et al., 2020; Fernandez-Beltran et al., 2021; Paudel et al., 2021, 2023b) without releasing the full datasets.

We present CY-Bench, a comprehensive dataset and benchmark for sub-national crop yield forecasting, covering thirty-eight countries for maize and twenty-nine countries for wheat across six continents. Here, sub-national refers to the administrative levels for which official crop statistics are published. Crop yield refers to the end-of-season yield reported in the statistics; and forecasting refers to the production of end-of-season yield estimates with a certain lead time before harvest (e.g., mid-season or 30 days before harvest). Thus, the dataset combines sub-national yield statistics with relevant predictors, such as growing-season

weather, remote sensing indicators, and soil properties. Coverage of diverse crop production systems enables a comprehensive evaluation of model performance across regions with heterogeneous agricultural practices and infrastructure, including low- and middle-income countries, which are generally under-represented in (machine learning) benchmarks. CY-Bench has been designed and curated by agricultural experts, climate scientists, and machine learning researchers from the AgML community (https://www.agml.org/), with the aim of facilitating model intercomparison across diverse agricultural systems around the

globe. By lowering the barrier to entry for machine learning researchers in this crucial application area, CY-Bench facilitates the development of improved crop forecasting tools that can be used to support decision-makers in food security planning worldwide.

## 2   Dataset construction

CY-Bench is a benchmark dataset to train and evaluate crop yield forecasting models that produce in-season forecasts with a

certain lead time ahead of harvest. The benchmark includes a comprehensive set of predictors that are known to be important drivers of crop yield. Crop yield is determined by the complex interaction of genetics (G), environmental conditions (E), and management decisions (M), commonly referred to as GxExM. Genetics (G) includes factors such as genotype, phenotype, and cultivar; environmental conditions (E) encompass both abiotic factors (e.g., climate, soil) and biotic factors (e.g., pests, pollinators); and management decisions (M) involve farm practices like tillage, sowing dates, irrigation, and fertilization (Liliane

and Charles, 2020). Technological advancements, including genetic improvements, better farm inputs, machinery, and enhanced management practices, have all contributed to increased yields over time (Liliane and Charles, 2020). These advancements are often reflected in the yield trend (Lecerf et al., 2019). At the sub-national spatial scale, data on genetic differences (e.g., cultivars) and farm management practices, with the exception of planting and harvest dates, are often not available. Therefore, previous studies of crop yield forecasting also commonly rely on environmental factors, crop calendar information (planting and harvest

dates), and yield trend (You et al., 2017; Khaki et al., 2020; Paudel et al., 2021). Factors not explicitly captured in CY-Bench but known to influence end-of-season yields are biotic stressors (e.g., pests and diseases), farm management choices (e.g., irrigation, fertilization, cultivar selection), and socioeconomic factors (e.g., market prices, labor availability, and policy changes).



## 2.1 Data sources selection

In the absence of benchmark datasets like CY-Bench, someone interested in modeling crop yield is faced with numerous
questions about data sources and quality. The data collection and pre-processing protocols of many predictor datasets, cropland
or crop type maps, crop calendars and official statistics are important to select suitable data sources. Each data source has
strengths and limitations related to spatial and temporal resolutions and coverage, sampling methods and gap-filling strategies.
After data source selection, predictor data from diverse spatial resolutions needs to be aggregated (to the level of yield statistics),
which can lead to information loss. In the process of constructing CY-Bench, we engaged a diverse community of researchers to
weigh the benefits and limitations of data sources for each type of data necessary to produce crop yield forecasts. The result of
this process is a set of data sources, including alternatives sources and our justifications for picking one data source over another.
Overall, data sources for CY-Bench were selected considering global coverage, public access, regular updates (except for static
data), near real-time availability and their relevance for crop growth and development.

### 2.1.1 Weather and soil moisture data

The most relevant weather variables for crop yield forecasting are temperature, solar radiation, and precipitation (Frieler et al.,
2017). Precipitation affects crop growth via soil moisture availability and evapotranspiration. Although actual evapotranspiration
would be preferred over potential evapotranspiration, the former is crop-dependent and not readily available at a global scale.

Temperature (`temp`), precipitation (`prec`), and radiation (`rad`) were selected from AgERA5 (Boogaard et al., 2022), which
provides daily data at a 0.1°($\sim$11 km) spatial resolution. Reference or potential evapotranspiration (`ETo`) was selected from the
FAO-AQUASTAT dataset (AQUASTAT, 2021), which relies on the FAO Penman-Monteith method (Allen et al., 1998) and takes
input variables from the AgERA5 dataset. AgERA5 offers agrometeorological indicators from 1979 to the present, derived from
ERA5 reanalysis and is tailored for agricultural studies. Its key benefits include high-quality data with near real-time updates (i.e.
lag of $\sim$2 weeks), comprehensive documentation, and free access via the Copernicus Climate Data Store (CDS). However, one
potential drawback is the absence of bias correction, which adjusts precipitation and temperature data using historical weather
observations to better reflect local conditions. In our case, where we are primarily interested in a sub-national scale, the absence
of bias correction is not considered a major concern.

For soil moisture, we selected surface soil moisture (`SSM`) and root-zone soil moisture (`RSM`) from the Global Land Data
Assimilation System (GLDAS) dataset (Rodell et al., 2004). This dataset represents gridded and global soil moisture data
developed by integrating satellite- and ground-based observational data products, using advanced land surface modeling and
data assimilation techniques. The dataset is available from 2003 to present, and can be freely downloaded from Goddard Earth
Sciences Data and Information Services Center (GES DISC). It has a temporal resolution of one day, and a spatial resolution of
0.25°($\sim$28 km). As an alternative, we considered GLEAM (Miralles et al., 2024). However, that dataset is typically updated
only once a year and is currently available only up to December 2023.



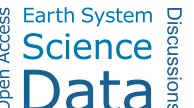

### 2.1.2 Remote sensing data

Remote sensing indicators of crop biomass and health include vegetation indices, such as the normalized difference vegetation index (`NDVI`) and enhanced vegetation index (`EVI`), as well as biophysical metrics like the fraction of absorbed photosynthetically active radiation (`fPAR`) and leaf area index (`LAI`). Sub-national yield forecasting requires long-term time series of these indicators, coupled with frequent satellite revisits to ensure cloud-free imagery. These requirements practically limit options to the coarse-resolution missions MODIS and its successor, VIIRS. While MODIS data can be directly downloaded

from NASA Land Processes Distributed Active Archive Center (LPDACC), the raw vegetation indices and biophysical variables are often of low quality due to issues like cloud cover. These limitations require further screening, gap-filling, and corrections. Furthermore, additional processing, which includes the use of quality flags and the application of temporal smoothing procedures, is time-consuming and complex. These challenges become even more pronounced when processing near-real-time data, which is essential for operational yield forecasting.

In view of an operational deployment of sub-national yield forecasting, we selected two analysis-ready operational products representing crop biomass and health: `fPAR` and `NDVI`. `fPAR` is provided as dekadal (10 day) data with a spatial resolution of 0.0045°(500 m), utilizing gap-filled and smoothed MODIS and VIIRS datasets. This data is sourced from EC-JRC, and its quality is being evaluated in Seguini et al. (In preparation). `NDVI`, a key indicator of vegetation greenness, is derived from MOD09CMG (Vermote, 2015), available from NASA LPDACC. The data is prepared as an eight-day composite with a spatial

resolution of 0.05°($\sim$ 5 km), selecting the pixel with the highest quality for each composite period. The quality of this `NDVI` product has been evaluated in Meroni et al. (2019).

### 2.1.3 Soil data

We selected data from the World Inventory of Soil Emission Potentials (WISE) project (Batjes, 2016) for static soil properties. WISE data is constructed using the soil map unit delineations of the broad-scale Harmonized World Soil Database, overlaid by a

climate zones map (Köppen-Geiger) as co-variate, and soil property estimates derived from analyses of the ISRIC-WISE soil profile database for the respective mapped 'soil/climate' combinations. The dataset has a spatial resolution of 30 arc-second (0.00833°($\sim$ 0.9 km)).

   While SoilGrids (Hengl et al., 2017; Poggio et al., 2021) is an alternative, WISE was selected due to its suitability for agricultural applications. Specifically, WISE data is considered to be more readily interpretable and provides essential parameters

like soil rooting depth and water holding capacity, which are absent in SoilGrids.

### 2.1.4 Crop mask and crop calendar data

Crop masks are selected from the European Space Agency WorldCereal (ESA WorldCereal) project (Van Tricht et al. (2023)), which provides an up-to-date and actively maintained source for cropland and crop type maps at a spatial resolution of 0.0045°(500 m). Alternative sources of crop masks include Anomaly Hotspots of Agricultural Production (ASAP) from EC-JRC

and IIASA (JRC-IIASA) and Global Best Available Crop Specific Masks (GEOGLAM-BACS) from the Group on Earth





Observations Global Agriculture Monitoring (GEOGLAM). We considered ESA WorldCereal to be a better choice than the generic cropland layer from JRC-IIASA because of the availability of crop type maps for maize and wheat (spring and winter cereals). Although GEOGLAM-BACS (Becker-Reshef et al., 2023) provides crop type maps for maize and wheat (spring and winter cereals), their spatial resolution (0.05°) is lower compared to ESA WorldCereal (0.00464°).

Crop calendars also come from the ESA WorldCereal project (Franch et al., 2022). ESA WorldCereal crop calendars combine information from existing global crop calendar products, such as GEOGLAM Crop Monitor, the United States Department of Agriculture Foreign Agricultural Service (USDA-FAS), FAO, and EC-JRC's ASAP, into a baseline map and sample them to train a Random Forest algorithm based on climatic and geographic data. They have global coverage and a spatial resolution of 0.5°(∼ 50 km). We considered alternative sources, including Food and Agriculture Organization (FAO), GGCMI (Waha

et al., 2012; Minoli et al., 2019), MIRCA (Portmann et al., 2010), and SAGE (Sacks et al., 2010). However, we selected ESA WorldCereal primarily due to its global coverage and alignment with our crop statistics data. A detailed comparison, based on crop types, country coverage, spatial resolution, and data sources, can be found in our GitHub Repository (Paudel et al., 2025b).

### 2.1.5  Crop statistics data

Crop yield statistics for sub-national administrative levels are obtained from national statistics offices or regional agencies,

depending on their quality and timely availability. In most cases, they come from the national statistics offices. For example, in the United States, they are published by the National Agricultural Statistics Service (NASS) of the United States Department of Agriculture (USDA). For the European Union, member countries report statistics to Eurostat. However, we considered Ronchetti et al. (2024) a more reliable source than Eurostat, as they follow a harmonization procedure developed by EC-JRC. For Germany, we selected data from Duden et al. (2024) instead of Ronchetti et al. (2024) because of better temporal coverage (1979-2021

vs 1999-2020), higher spatial resolution for maize (NUTS level 3 vs level 1) and better quality based on consistency checks (e.g., $yield = production/area$). For Africa, except for Mali, data comes from the Famine Early Warning Systems Network (FEWS NET) Data Warehouse. The data was compiled and harmonized to account for changing administrative boundaries by Lee et al. (Under review). For Mali, we selected Compagnie Malienne pour le Developpement des Textiles (CMDT) dataset (ICRISAT Mali, 2018) that provides higher spatial resolution data at arrondissement-level (administrative level 3). Depending

on the country, the term 'sub-national' can refer to administrative division 1 (province, state, region), division 2 (district), or division 3 (county, municipality, commune) (Table 2). When statistics for multiple administrative levels are available, we select the highest resolution.

### 2.2  Data preparation

### 2.2.1  Crop yield data (targets)

CY-Bench dataset includes crop statistics from thirty-eight countries for maize and twenty-nine countries for wheat (Figures 3 and 4). Coverage maps show that CY-Bench has extensive coverage when layered on top of crop type maps from ESA WorldCereal, with notable omissions including Canada, Ukraine and Russia for wheat and Ukraine, Uganda and Tanzania





for maize. Data preparation for yield data involved filtering out values that do not meet certain consistency checks, e.g., $yield \neq production/area$, or zero values. The data sources or publications from which CY-Bench draws the data do additional
consistency checks. We refer interested readers to respective data cards Pushkarna et al. (2022) in our GitHub repository (Paudel et al., 2025b) which contains further links to data sources, related reports and publications.

### 2.2.2 Predictor data

CY-Bench predictor data includes static soil properties and time series of weather variables, soil moisture indicators and vegetation indicators (Table 1). Predictor data and yield statistics often differ in spatial and temporal resolution, requiring further
processing to align them effectively. Weather, `ETo` and soil moisture data come in daily time steps. `fPAR` comes in dekadal time step, with three values per month (days 1-10, 11-20, 21-31). `NDVI` data is available every eight days, with gaps due to cloud cover.

Predictor data is filtered using crop type maps (or crop masks) from EC-JRC (2024) which are derived from the ESA WorldCereal project (Van Tricht et al., 2023). This step restricts predictor data to pixels in harvested crop areas only. After
masking, predictor data is aggregated to match the boundaries and spatial level of the yield data according to the administrative level (Figure 1). The data preparation workflow is implemented in a Python script in our GitHub repository (Paudel et al., 2025b). We note that all predictor data retain their temporal resolution from the original data source, creating a multi-modal dataset.

### 2.2.3 Additional pre-processing for yield forecasting models

Here we describe some additional pre-processing implemented in our `cybench` library that are relevant for building crop
yield forecasting models. Predictor data from different sources come with different temporal coverage. Similarly, they include observations for the calendar year, which may not capture the crop season. First, we align time series inputs (weather variables, remote sensing indicators and soil moisture indicators) to the crop season (see Figure 2). We define the boundaries of the crop season as 90 days before the start of season (the spin-up time) to the end of season in a particular calendar year and filter out data outside the boundaries. Therefore, data from the previous year can be included in the current calendar year's crop season
and data after the end-of-season date get pushed to the crop season for the next calendar year. Furthermore, data towards the end-of-season are filtered out based on the lead time relative to harvest or end-of-season. Second, we align the input data sources and label data to produce a set of data samples that are complete, i.e. each data sample includes all the relevant predictors for each time step (or static) and a label.

The time series predictors need further pre-processing during modeling. Certain models require time series data to have the
same number of time steps. Therefore, time series inputs are aggregated to dekadal time steps (days 1-10, 11-20, 21-30, and so on), taking the mean of most variables, minimum of minimum temperature, maximum of maximum temperature and the sum of precipitation flux, climatic water balance and solar radiation flux. Where the variable is categorical (such as soil drainage), we take the mode.

To further prepare features as tabular data, time series data are aggregated in the temporal dimension to create domain-relevant
features. Following expert recommendations we create monthly averages of minimum daily temperature (`tmin`), maximum



**Table 1.** Overview of the predictor data, crop mask and crop calendar

Abbreviations: Temperature (`temp`), Precipitation flux (`prec`), Solar radiation flux (rad), Potential evapotranspiration (`ETo`), Climatic water balance (`CWB`), Fraction of absorbed photosynthetically active radiation (`fPAR`), Normalized difference vegetation index (`NDVI`), and available water capacity (`AWC`).

| Category | Data | | Spatial resolution | Temporal resolution | Source |
|---|---|---|---|---|---|
| | Name | Unit | | | |
| Weather (time series) | temp | °C | 0.1°(11 km) | daily | AgERA5 (Boogaard et al., 2022) |
| | prec | mm | | | |
| | rad | J m$^{-2}$ | | | |
| | ETo | mm | 0.1°(11 km) | daily | AQUASTAT-FAO (AQUASTAT, 2021) |
| | CWB | mm | 0.1°(11 km) | daily | Computed as prec - ETo |
| | soil moisture | kg m$^{-2}$ | 0.25°(28 km) | daily | NASA GLDAS (Rodell et al., 2004) |
| Vegetation (time series) | fPAR | % | 0.0045°(0.5 km) | 10 days | JRC (Seguini et al., In preparation) |
| | NDVI | - | 0.05° (5.5 km) | 8 days | MOD09CMG (Vermote, 2015) |
| Soil (static) | AWC | cm m$^{-1}$ | 0.0083°(0.9 km) | static | WISE (Batjes, 2016) |
| | bulk density | kg dm$^{-3}$ | | | |
| | drainage class | - | | | |
| Crop (auxiliary) | crop mask | % | 0.0045°(0.5 km) | static | WorldCereal (Van Tricht et al., 2023; EC-JRC, 2024) |
| | crop calendar | day | 0.5° (55 km) | static | WorldCereal (Franch et al., 2022) |

daily temperature (`tmax`), average daily temperature, daily precipitation (`prec`), cumulative climatic water balance (`prec - ETo`) and surface soil moisture. Similarly, monthly maximum values are calculated for cumulative growing degree days (`GDD`), cumulative precipitation, cumulative `fPAR` and cumulative `NDVI`. Furthermore, we calculate the number of days in which `tmin` is less than 0°C ('cold days'), days in which `tmax` is greater than 35°C ('hot days') and days where `prec` is less than 1
mm ('dry days').

### 2.2.4 Future expansion and data integration

CY-Bench currently includes predictor data up to and including 2023. Availability of crop statistics varies by country (see Figures 5, 6). We share yield and predictor data preparation scripts and notebooks in our GitHub repository (Paudel et al., 2025b) to make the inclusion of new data possible as it becomes available. For example, when crop statistics for 2024 become
available for specific countries, the data preparation pipeline for agricultural yield data can be run for the crop statistics, and predictor data preparation scripts can be run for predictor inputs. Expanding the database in the future primarily depends on onboarding crop statistics, as the global availability of input predictors ensures that integrating additional crop statistics is the only prerequisite for extending CY-Bench's coverage.





**Figure 1.** Overview of the CY-Bench data preparation process.

# 3 Dataset and task summary

## 3.1 Dataset overview

CY-Bench covers two main crops, namely maize and wheat. Depending on the country, the crop names can refer to different varieties or seasons of maize and wheat as detailed in our GitHub repository (Paudel et al., 2025b).



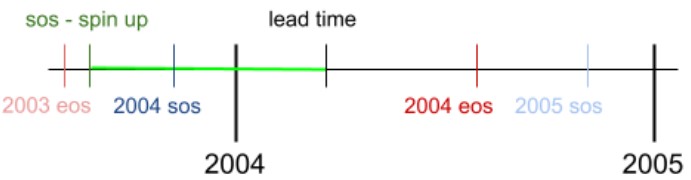

**Figure 2.** Alignment of time series predictors to the crop season.

**Table 2.** Countries and the administrative levels for which CY-Bench data is provided

| Group | Country name (country code) : Admin level or NUTS Level | | |
|---|---|---|---|
| EU (n=23) | Austria (AT) : 2 | Belgium (BE) : 2 | Bulgaria (BG) : 2 |
| | Czechia (CZ) : 3 | Germany (DE): 3 | Denmark (DK) : 3 |
| | Estonia (EE) : 3 | Greece (EL) : 3 | Spain (ES) : 3 |
| | Finland (FI) : 3 | France (FR) : 3 | Croatia (HR) : 2 |
| | Hungary (HU) : 3 | Ireland (IE) : 2 | Italy (IT) : 3 |
| | Lithuania (LT) : 3 | Latvia (LV) : 3 | Netherlands (NL) : 2 |
| | Poland (PL) : 2 | Portugal (PT) : 2 | Romania (RO) : 3 |
| | Sweden (SE) : 3 | Slovakia (SK) : 3 | |
| FEWSNET (n=12) | Angola (AO) : 1 | Burkina Faso (BF) : 2 | Ethiopia (ET) : 2 |
| | Lesotho (LS) : 1 | Madagascar (MG) : 2 | Malawi (MW) : 2 |
| | Mozambique (MZ) : 1 | Niger (NE) : 2 | Senegal (SN) : 2 |
| | Chad (TD) : 1 | South Africa (ZA) : 1 | Zambia (ZM) : 2 |
| Other countries (n=8) | Argentina (AR) : department | Australia (AU) : ABARES region (sub-state) | |
| | Brazil (BR) : municipality | China (CN) : province | |
| | India (IN) : district | Mali (ML) : Municipality | |
| | Mexico (MX) : state | United States (US) : county | |

## 3.2 Task

CY-Bench is designed to train and evaluate models for in-season crop yield forecasting of wheat and maize at the sub-national

level, covering major and underrepresented crop-growing countries worldwide. Forecasts can be made at multiple time points from start of season (sos) to end of season (eos), based on a lead time relative to eos, e.g., middle-of-season $((\text{eos} - \text{sos})/2)$, quarter-of-season $((\text{eos} - \text{sos})/4)$ and $n$-days before harvest. The exact inference time depends on the crop calendar for the selected crop and region. The quarter-of-season mark often coincides with crops reaching physiological maturity, while the middle-of-season typically represents the transition from vegetative to reproductive growth stages (Lee et al., 2022; Basso and

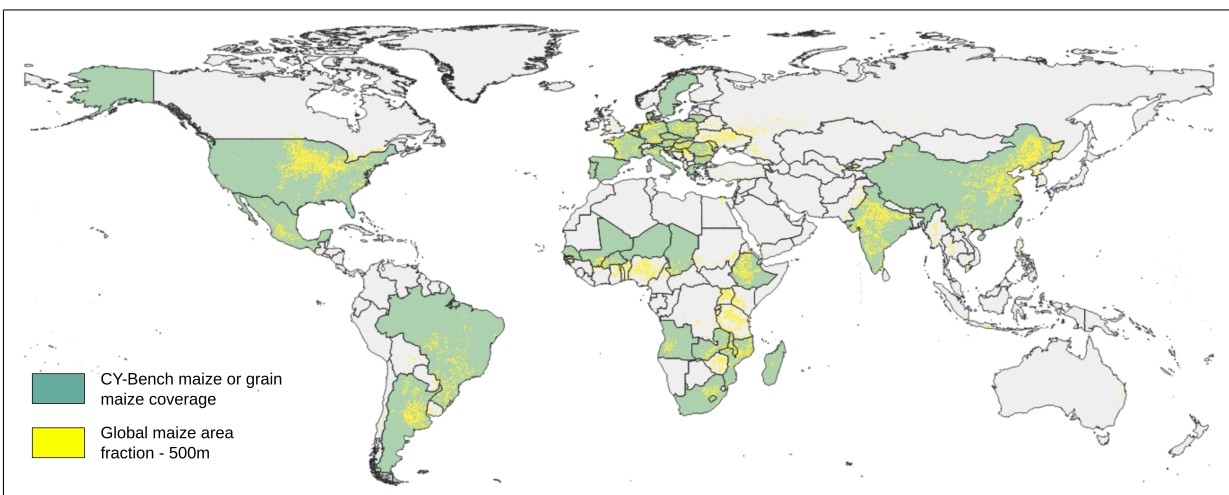

**Figure 3.** A map of the countries covered by CY-Bench for maize yield forecasting. CY-Bench covers 38 countries in total.

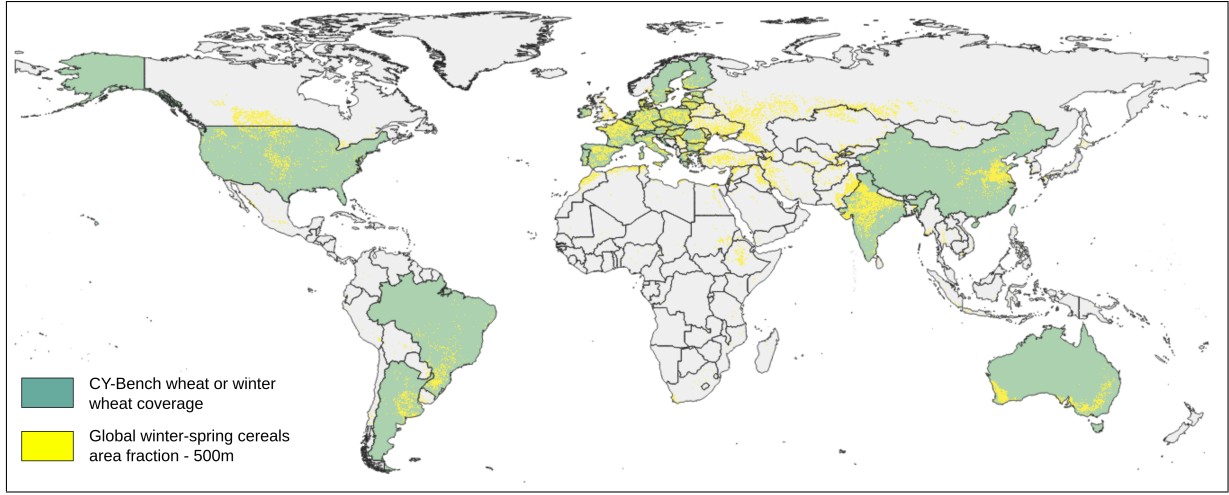

**Figure 4.** A map of countries covered by CY-Bench for wheat yield forecasting. CY-Bench covers 29 countries in total.

Liu, 2019). The reproductive period, which includes flowering and grain filling, is generally the most climate-sensitive phase of the growing season (Hatfield and Prueger, 2015). By contrast, the vegetative phase before mid-season and the senescence phase after grain filling tend to be less sensitive to climate anomalies, although rainfall during the harvest period can significantly impact yield. While quarter-of-season forecasts may achieve higher accuracy, middle-of-season forecasts balance accuracy with actionable insights, allowing for management adjustments during the remainder of the growing season—something late

forecasts cannot provide.



| Country | ADMIN SIZE km² | Maize | | | | Wheat | | | |
| | | YEARS MIN-MAX | ADMIN REGIONS | LABELS | LABELS TIMELINE 2003-2024 | YEARS MIN-MAX | ADMIN REGIONS | LABELS | LABELS TIMELINE 2003-2024 |
| --- | --- | --- | --- | --- | --- | --- | --- | --- | --- |
| Austria | 9541 | 2004-2020 | 9 | 153 | | 2004-2020 | 9 | 133 | |
| Belgium | 3011 | 2011-2020 | 10 | 85 | | 2004-2020 | 11 | 174 | |
| Bulgaria | 19435 | 2004-2020 | 6 | 102 | | 2010-2020 | 6 | 45 | |
| Czechia | 5300 | 2005-2020 | 14 | 212 | | 2004-2020 | 14 | 238 | |
| Germany | 798 | 2004-2021 | 215 | 3162 | | 2004-2021 | 358 | 5888 | |
| Denmark | 3496 | 2011-2020 | 3 | 22 | | 2006-2020 | 10 | 150 | |
| Estonia | 9065 | - | 0 | 0 | | 2004-2020 | 5 | 85 | |
| Greece | 2345 | 2009-2019 | 40 | 440 | | 2003-2019 | 40 | 609 | |
| Spain | 8039 | 2003-2020 | 50 | 830 | | 2003-2020 | 43 | 536 | |
| Finland | 10626 | - | 0 | 0 | | 2004-2020 | 18 | 176 | |
| France | 5976 | 2004-2020 | 92 | 1539 | | 2003-2020 | 91 | 1531 | |
| Croatia | 28211 | 2005-2020 | 2 | 32 | | 2008-2020 | 2 | 25 | |
| Hungary | 4394 | 2004-2020 | 20 | 340 | | 2004-2020 | 20 | 278 | |
| Ireland | 25740 | - | 0 | 0 | | 2010-2020 | 3 | 11 | |
| Italy | 2476 | 2003-2020 | 101 | 1592 | | 2003-2020 | 83 | 1197 | |
| Lithuania | 6306 | 2004-2020 | 10 | 140 | | 2004-2020 | 10 | 169 | |
| Latvia | 12188 | - | 0 | 0 | | 2004-2018 | 5 | 75 | |
| Netherlands | 2960 | 2008-2020 | 12 | 126 | | 2004-2020 | 12 | 195 | |
| Poland | 18177 | 2004-2020 | 17 | 284 | | 2004-2020 | 17 | 284 | |
| Portugal | 4968 | 2003-2020 | 5 | 88 | | 2004-2020 | 4 | 68 | |
| Romania | 5547 | 2003-2020 | 42 | 705 | | 2004-2020 | 34 | 379 | |
| Sweden | 11671 | 2007-2020 | 1 | 10 | | 2004-2020 | 17 | 271 | |
| Slovakia | 6538 | 2007-2018 | 8 | 94 | | 2017-2018 | 5 | 10 | |

**Figure 5.** Data size summary per dataset for EU countries





| Country | ADMIN SIZE km² | Maize | | | | Wheat | | | |
|---|---|---|---|---|---|---|---|---|---|
| | | YEARS MIN-MAX | ADMIN REGIONS | LABELS | LABELS TIMELINE 2003-2024 | YEARS MIN-MAX | ADMIN REGIONS | LABELS | LABELS TIMELINE 2003-2024 |
| Australia | 123694 | - | 0 | 0 | | 2003-2022 | 17 | 270 | |
| China | 166114 | 2003-2022 | 31 | 595 | | 2004-2022 | 25 | 475 | |
| India | 3950 | 2003-2017 | 498 | 6498 | | 2004-2017 | 474 | 6261 | |
| Angola | 59858 | 2004-2017 | 17 | 238 | | - | 0 | 0 | |
| Burkina Faso | 5392 | 2003-2019 | 45 | 540 | | - | 0 | 0 | |
| Ethiopia | 10350 | 2003-2020 | 60 | 722 | | - | 0 | 0 | |
| Lesotho | 2864 | 2004-2021 | 10 | 163 | | - | 0 | 0 | |
| Madagascar | 21902 | 2005-2010 | 22 | 132 | | - | 0 | 0 | |
| Mali | 3183 | 2003-2017 | 24 | 360 | | - | 0 | 0 | |
| Malawi | 3117 | 2018-2023 | 4 | 16 | | - | 0 | 0 | |
| Mozambique | 75397 | 2004-2022 | 10 | 159 | | - | 0 | 0 | |
| Niger | 4404 | 2003-2021 | 25 | 264 | | - | 0 | 0 | |
| Senegal | 2853 | 2003-2015 | 40 | 401 | | - | 0 | 0 | |
| Chad | 40330 | 2003-2017 | 17 | 231 | | - | 0 | 0 | |
| South Africa | 123172 | 2004-2022 | 9 | 167 | | - | 0 | 0 | |
| Zambia | 8892 | 2004-2017 | 71 | 994 | | - | 0 | 0 | |
| Mexico | 58682 | 2014-2022 | 64 | 133 | | - | 0 | 0 | |
| United States | 1614 | 2003-2023 | 1938 | 32162 | | 2004-2023 | 1638 | 22834 | |
| Argentina | 3174 | 2003-2023 | 298 | 5582 | | 2003-2024 | 238 | 4607 | |
| Brazil | 423 | 2003-2023 | 4483 | 87421 | | 2003-2022 | 1015 | 18429 | |

**Figure 6.** Data size summary per dataset for other (non-EU) countries





### 3.2.1 Formal definition

The input data consists of time series inputs (weather, soil moisture, and vegetation indices) and static inputs (soil properties). Let $\mathbf{x}_t$ represent the vector of time series inputs at time $t$, where $t$ spans from $\texttt{sos}$ up to the inference point $T$. Time series data up to the inference point is represented as $\mathbf{X}_{\text{sos}:T} = (\mathbf{x}_{\text{sos}}, \mathbf{x}_{\text{sos}+1}, \ldots, \mathbf{x}_T)$ and static inputs as $\mathbf{z}$. Each training or testing sample $i$ corresponds to a specific region-season pair $(r, s)$. For each training sample $i = (r, s)$, the input consists of $\mathbf{X}_{\text{sos}:T}^{(i)}$ and $\mathbf{z}^{(i)}$. The target is the end-of-season yield $Y^{(i)}$ for the corresponding region $r$ and season $s$. The objective is to learn a mapping function $f$ such that $Y^{(i)} = f(\mathbf{X}_{\text{sos}:T}^{(i)}, \mathbf{z}^{(i)}; \theta) + \epsilon^{(i)}$, where $\theta$ represents the model parameters, and $\epsilon^{(i)}$ is the error term.

During testing, the model gets $\mathbf{X}_{\text{sos}:T}^{(j)}$ from the start of the season ($\texttt{sos}$) up to the inference point $T$ and static inputs $\mathbf{z}^{(j)}$ for a new sample $j = (r', s')$. The model then forecasts the end-of-season yield $\hat{Y}^{(j)} = f(\mathbf{X}_{\text{sos}:T}^{(j)}, \mathbf{z}^{(j)}; \hat{\theta})$, where $\hat{\theta}$ are the model parameters learned during training. Model performance is evaluated by comparing yield forecasts $\hat{Y}^{(j)}$ with reported yields $Y^{(j)}$.

Some details that are ignored in the above formulation:

- $t$ can actually start earlier than $\texttt{sos}$, based on spin-up time (e.g., 60 days or 90 days before $\texttt{sos}$).

- The temporal resolution can be different for time series inputs, i.e. the $+1$ in $\texttt{sos} + 1$ represents different time units.

## 3.3 Training, validation and test splits

We closely mimic real-world operational settings in the predictor data used, data pre-processing steps and evaluation setup. Specifically, we adopt Leave-One-Year-Out validation (LOYO), which is more appropriate for this type of agricultural problem compared to the random sampling methods commonly used in prior studies (Richetti et al., 2023), including SustainBench (Yeh et al., 2021). Our library computes the following evaluation metrics: normalized root mean squared error (NRMSE; i.e., the root mean squared error normalized by the average yield of the test set), mean absolute percentage error (MAPE), and R-squared ($R^2$). These metrics are computed by averaging over all cross-validation test folds (which covers the complete dataset for LOYO) and all admin regions with a country.

## 4 Discussion

### 4.1 Impact

In addition to the relevance for climate change, food security and the United Nations' sustainable development goals, CY-Bench dataset is relevant to the earth systems science and machine learning research communities due to its comprehensive geographic coverage, capturing diverse agricultural practices and conditions. The inclusion of (indicators derived from) satellite imagery, weather data, and soil properties provides a rich, heterogeneous dataset that presents numerous opportunities for the development of innovative machine learning methods. An inherent challenge of agricultural data, and crop-yield forecasting specifically, is the high level of domain knowledge required in collecting and processing the various data types and defining the task. This



analysis-ready dataset is accessible to modelers who do not necessarily have expertise in yield forecasting, lowering the barrier to entry for advanced yield forecasting research and fostering broader participation and innovation in the field. Beyond academic research, this dataset can significantly impact policy-making, agricultural planning, and disaster response by enabling the robust evaluation and development of operational models. Researchers, policymakers, farmers, and agribusinesses can benefit from the insights derived from this dataset, leading to better-informed decisions and improved agricultural outcomes.

## 4.2 Limitations

We also would like to point out several limitations of CY-Bench that can also be areas for improvement in future iterations.

1. Some limitations stem from the data sources available in the public domain.

   – The predictors do not capture certain factors that influence end-of-season yields, such as pests, diseases, and farm management choices. Similarly, CY-Bench excludes socioeconomic factors such as market prices, labor availability, and policy changes. Some of these omitted factors, like crop varieties and management practices, might exhibit spatial or temporal correlations. Consequently, these factors could be partially captured by incorporating spatial or temporal embeddings within a model. Other factors, such as pests, are less likely to be adequately captured in this way. Including these variables could provide a more holistic understanding of yield fluctuations and help develop more robust models. Nevertheless, the availability of such data on a global scale is typically difficult or often not feasible.

   – The crop masks and crop calendars included in CY-Bench are static, i.e. they do not reflect yearly changes. Access to dynamic and up-to-date masks and calendars may improve the performance of yield forecasting models. Similarly, crop calendar information could be enriched with data on phenology transitions. Information on phenology changes could improve expertise-based feature design and produce more predictive features.

   – Crop yield forecasting models could benefit from incorporating weather forecasts (Cunha et al., 2018). In our task definition, models cannot access data after the lead time and, therefore, cannot capture conditions that might affect the end-of-season yields after that point. In the real-life setting, forecasters would have access to weather forecasts that may provide useful information. We did not include weather forecasts in CY-Bench because the evidence supporting their usefulness is variable (Darbyshire et al., 2020) and there are multiple considerations that need to be discussed before including weather forecasts: a) Observed weather data and forecast data may originate from difference sources. b) Some variables, e.g., `fPAR` and `NDVI`, have no forecasts. Strategies for addressing these gaps are necessary. c) Weather forecasts from General Circulation Models (GCMs) typically have coarse spatial resolutions (50-400 km grid sizes) and often contain systematic errors or biases that must be adjusted. Bias correction and downscaling techniques must be applied using observed historical records of weather variables, such as precipitation and temperature, to adjust the climate data and better represent local conditions. d) Using weather forecasts for yield prediction would also cause error/uncertainty propagation from the weather forecast models leading to an increase in overall uncertainty.





2. CY-Bench does not differentiate between irrigated and non-irrigated systems. These systems can exhibit different responses to predictors due to varying water availability, leading to potential inaccuracies in yield forecasts. Our choice was driven by the fact that crop statistics in most countries are rarely reported separately for irrigated and non-irrigated areas.

3. CY-Bench does not provide process-based crop model outputs, which could serve as valuable input features for machine learning models. Additionally, the current feature aggregation uses fixed time steps rather than adapting to crop growth stages. Access to crop model outputs, which contain information on key phenological state changes, could enable the development of more effective, stage-specific features.

4. CY-Bench does not provide raw surface reflectance but includes vegetation indices (i.e. `NDVI` – the most frequently-used index for crop yield forecasting (Schauberger et al., 2020), and `fPAR`), that are strongly correlated with yields (Johnson, 2016). The utility of high-resolution (10-60 m) satellite images (considering spatial information without aggregating to administrative units as performed in CY-Bench) for crop yield forecasting at the sub-national level will result in a massive data size. In addition, the temporal availability of such data is not very long (e.g., Sentinel available since 2014/15). Meanwhile, moderate-resolution satellite images have been used to forecast crop yields in the United States (e.g., You et al. (2017)), demonstrating an advantage over time series data. Future work could investigate the value of satellite images for sub-national crop yield forecasting at a global level.

5. Finally, the LOYO method of evaluation is used due to small data sizes in many countries. This approach assumes that all years are independent, which may be too strong of an assumption if consecutive years have correlated environmental and climatic conditions. Also, management strategies might affect multiple seasons; for example, there is evidence of the impact of crop rotation from a previous season on yields Lawes et al. (2022).

### 4.3 Advancing earth system modeling

We noted a distinct lack of benchmark datasets for agricultural yield forecasting. Still, many recent developments in the related field of crop type mapping using satellite data (Rußwurm et al., 2019; Tseng et al., 2021b; Yeh et al., 2021; Kondmann et al., 2021) are leading to exciting progress in the development of methods for extracting meaningful patterns from time series of earth observation data (Rußwurm and Körner, 2018; Rußwurm et al., 2019; Pelletier et al., 2019; Sainte Fare Garnot et al., 2020). Similarly, some studies have shown improved model performance for land cover classification, crop mapping and agricultural yield forecasting using meta-learning and multitask learning (Tseng et al., 2022, 2021a; Kerner et al., 2020). CY-Bench includes time series of crop productivity or vegetation health indicators from earth observation as predictors, and can therefore be combined with existing crop mapping benchmark datasets to explore such approaches.

Apart from the downstream task of pre-harvest yield forecasting, CY-Bench enables explorations in transfer learning, domain adaptation, and representation learning to assess whether models can generalize well across diverse geographic and climatic conditions. We envision at least four directions for future research. First, transfer learning methods can be explored to improve model generalization ability when training on data-rich regions and deploying to data-sparse regions (Koukos et al., 2024; Coulibaly et al., 2019; Nowakowski et al., 2021). Second, self-supervised learning could be used to harness the vast amounts of

unlabeled agricultural data available (Wang et al., 2022; Xu et al., 2024). By training models to recognize patterns and structures within this data, we can build robust representations that capture essential features of agricultural systems. These representations can then be fine-tuned using the labeled datasets in CY-Bench specific to each country or crop. For instance, a self-supervised model trained on satellite images and environmental data can later be fine-tuned to predict specific crop yields in various regions, making it a powerful tool for global agricultural analysis. Third, another important area is to explore the stability of model predictions against natural and human interventions. This involves understanding how factors like extreme weather events, policy changes, or management practices impact yield forecasts. Causal invariance learning focuses on identifying and utilizing stable variables across different environments to ensure robustness and generalization (Mitrovic et al., 2020; Neophytides et al., 2024). For example, soil quality and basic climatic factors like temperature and precipitation may have stable relationships with crop yields. By recognizing variables that consistently impact crop yields regardless of geographic or climatic differences, it may be possible to build models that are resilient to distributional shifts and perform reliably across diverse conditions.

## 5 Conclusions

Innovative, data-driven approaches are crucial for enhancing the resilience of food systems to climate change and extreme events, which is essential for achieving the United Nations' Sustainable Development Goal 2 of Zero Hunger (Schneider et al., 2023b). By providing a well-curated dataset designed for the consistent development and evaluation of large-scale crop yield predictions, CY-Bench is a step forward in improving the accuracy of yield forecasting. Curated by an interdisciplinary group of experts in agronomy, food security, climate science and agriculture, this dataset can facilitate increased collaboration between fields and ultimately help to produce reliable crop yield forecasts to support the decisions of farmers, policymakers and commodity traders worldwide. In summary, with CY-Bench we aim to contribute to advancing earth system sciences by providing a critical resource for modeling the effects of climate change, extreme events, and environmental variability on crop yields, while also supporting machine learning research in time series forecasting, transfer learning, and domain adaptation techniques.

## 6 Code and data availability

The complete code base encompassing data pre-processing, tools for model construction, training, evaluation, and data/-metric visualization routines is available through our publicly accessible GitHub repository: https://github.com/WUR-AI/AgML-CY-Bench/ (Paudel et al., 2025b). A summarizing overview can be found on https://cybench.agml.org/. We additionally provide a Python package `cybench` that can be installed via the repository to load the dataset and run CY-Bench. The dataset is available in Zenodo at https://doi.org/10.5281/zenodo.11502142 (Paudel et al., 2025a) and is comprehensively documented using the framework of Data Cards. Each individual dataset subset is accompanied by a dedicated Data Card located within the data_preparation directory of our repository (Paudel et al., 2025b). The CY-Bench dataset and the python library are licensed under EUPL-1.2, which is compatible with all of the licenses for the datasets included.

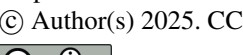



## Appendix A

*Author contributions.* DP: Conceptualization, Methodology, Project administration, Investigation, Software, Writing - original draft preparation, Writing - review & editing

MK: Project administration, Software, Validation, Writing - original draft preparation, Writing - review & editing

SOA: Data curation, Software, Validation, Writing - original draft preparation, Writing - review & editing

HB: Investigation, Software, Visualization, Writing - review & editing

RVB: Software, Validation, Writing - review & editing

AP: Software, Validation, Writing - review & editing

PP: Data curation, Validation, Writing - original draft preparation, Writing - review & editing

AS: Data curation, Validation, Writing - review & editing

WA: Data curation, Validation, Writing - original draft preparation, Writing - review & editing

MVB: Validation, Writing - original draft preparation, Writing - review & editing

AC: Validation, Writing - review & editing

OE: Data curation, Validation, Writing - original draft preparation, Writing - review & editing

RH: Data curation, Project administration, Validation, Writing - review & editing

RL: Data curation, Validation, Writing - review & editing

DL: Data curation, Validation, Writing - review & editing

IL: Data curation

MM: Data curation, Validation, Writing - original draft preparation, Writing - review & editing

JMM: Data curation, Writing - review & editing

SM: Data curation, Validation, Writing - original draft preparation, Writing - review & editing

JR: Data curation, Writing - original draft preparation, Writing - review & editing

ACR: Validation, Writing - original draft preparation, Writing - review & editing

RS: Data curation, Validation, Writing - original draft preparation, Writing - review & editing

GS: Data curation, Validation, Writing - review & editing

VS: Validation, Writing - original draft preparation, Writing - review & editing

RDSNJ: Data curation, Validation, Writing - review & editing

AKS: Data curation, Validation, Writing - original draft preparation, Writing - review & editing

RS: Writing - original draft preparation, Manuscript writing and review

LS: Validation, Writing - original draft preparation, Writing - review & editing

PV: Data curation, Validation, Writing - review & editing

INA: Funding acquisition, Supervision, Validation, Writing - review & editing

*Competing interests.* The authors declare that they have no conflict of interest.



*Acknowledgements.* CY-Bench benefited from many helpful discussions with the participants of AgML, the Machine Learning team of the Agricultural Model Intercomparison and Improvement Project (AgMIP).

We would also like to acknowledge the contributions of Marc Russwurm, Afef Marzougui, Hendrik Boogaard, Marijn van der Velde, Steven Hoek, Filip Szabo, Francesco Collivignarelli, Xiaomao Lin, Toshi Iizumi, Peng Fu, Prakriti Bista, Paresh Shirsath, Soora Naresh Kumar, Sibiri Traore and Javier Garcia Navarro in the design and implementation of the benchmark and preparation of the manuscript.

This work was partially supported by the WUR Research Investment Theme on Data-Driven Discoveries in a Changing Climate, and the Digital Europe Programme under Grant agreement AgrifoodTEF - Test and Experiment Facilities for the Agri-Food Domain (ID 101100622).





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
