# Peer review of "CY-Bench: A comprehensive benchmark dataset for sub-national crop yield forecasting"

_Earth System Science Data, 2025_

## Author Comment (AC2)

**Response to Reviewers**

Article Ref.: essd-2025-83

**CY-Bench: A comprehensive benchmark dataset for sub-national crop yield forecasting**

Earth System Science Data

Dear Editor and Reviewers,

We thank you for the insightful suggestions, feedback, and the time you have dedicated to review our manuscript. This is our initial response to your comments. We have provided a point-by-point response below, addressing each specific comment and suggestion.

We appreciate your consideration of the effort we made. Please contact us if there are other further suggestions.

Thank you for your kind consideration.

Sincerely,

Authors

**Reviewer #1 comments and suggestions**:

This study proposed a crop yield prediction benchmark for sub-national crop yield forecasting. The authors did a good job to build the dataset and the data sharing platform. However, there are some fundamental issues with the study design and protocols that need to be addressed before it can be considered for publication.

Thank you for acknowledging the effort dedicated to preparing the dataset and developing the data-sharing platform. We have addressed the concerns regarding the study design and protocols, and the specific revisions are outlined in the point-by-point responses below.

**Novelty:** The novelty of this study is not entirely clear. Based on the reviewer's previous experience reviewing/reading benchmark dataset papers, researchers typically contribute either newly collected datasets—often the result of years of fieldwork or labor-intensive data labeling—or propose novel modeling approaches that produce grid-level maps with global coverage. Importantly, such studies also provide new insights derived from their unique datasets. In contrast, the current study relies on publicly available yield statistics, processes commonly used data sources, and generates predictors following standard protocols. Both the yield data and

predictors have already been widely used in previous research. While compiling them in one place is useful, it is difficult to identify what is truly novel about this work.

The novelty in this study is rooted in the transdisciplinary approach to building a unified and comprehensive dataset that effectively integrates previously disparate components which have previously impeded similar applications. Our work directly responds to key gaps and priority needs articulated by the Machine Learning team of the Agricultural Model Intercomparison and Improvement Project (AgMIP). We expand on this effort's novelty in several key aspects:

- Benchmarking: Unlike existing research, which typically relies on single a country or sub-divisions within a country, our study provides a comprehensive, publicly available benchmark dataset, enabling consistent comparisons across models and approaches in several regions. This contrasts with the current landscape where datasets are often fragmented across different studies, and applies a non-standardized data processing steps hindering their potential as benchmarks. While the underlying data draws from existing public sources, its novelty lies in the integration and standardization across multiple countries (something not available in a single open-access source). The relevance of this effort is also underscored by recent literature. For example, Davis et al. (2025) highlight the urgent need for comprehensive and standardized subnational agricultural production datasets, while Sweet et al. (2025) specifically call for coordinated, interdisciplinary benchmark initiatives in agricultural modeling.

- Harmonization: We have put considerable effort into harmonizing the datasets, focusing on aligning them spatially and temporally, while also applying unit conversions across all sources. This ensures consistency across diverse data sources and provides an analysis-ready resource, eliminating the need for extensive preprocessing.

- Broad geographical coverage: Our study spans multiple continents, including regions with diverse agricultural practices and infrastructure, particularly low- and middle-income countries that are often underrepresented in (machine learning) benchmarks. This broad scope not only enables the development of robust models but also offers the potential to evaluate them across a wide range of crop production systems and contexts, reinforcing better generalization and addressing the need for research in under-served areas.

- Panel of experts: The dataset has been developed with input from a diverse group of experts in crop yield prediction, ensuring that it reflects a consensus on the most relevant variables and their source selection.

*We will revise our manuscript to more explicitly emphasize the significance of our effort and our contributions as outlined above*

**Static crop masks:** Crop masks play a crucial role in processing yield predictors, as they help eliminate irrelevant pixels and reduce noise in the data. However, this study applies a static

crop mask across multiple years, which is problematic. This approach does not align with the standards typically expected in benchmarking studies.

We acknowledge that dynamic crop masks could improve the accuracy of yield predictions by better reflecting spatial-temporal variability, as we also mentioned in the limitations section of the manuscript.

However, the availability of consistent, high-quality, dynamic crop masks at the global scale remains limited — a constraint shared by many yield prediction and benchmarking efforts (Kebede et al., 2025). To the best of our knowledge, several established benchmark datasets similarly rely on static or infrequently updated crop masks, as dynamic alternatives are often unavailable for large parts of the world (Yeh et al., 2021).

That said, CY-Bench is designed as a living resource that will evolve as the research community develops or identifies improved datasets including higher-quality crop masks, thematic predictors, and expert-curated features that can enhance subnational yield prediction.

Looking ahead, two options exist for incorporating dynamic crop masks:

1. regional, high-resolution products — such as the USDA Cropland Data Layer for the U.S. (Boryan et al., 2011) which would improve local accuracy but would also break the harmonization of the dataset, as these region-specific masks would not be applicable globally. This presents a challenge, as one of our goals is to enable fair intercomparisons between models across diverse regions.

2. the very recently released MIRCA-OS crop dataset (Kebede et al., 2025), which provides global crop masks at 5-year intervals. While this resource offers an exciting opportunity, its relatively coarse spatial resolution and recency mean we will first evaluate its suitability before integrating it into CY-Bench.

*We will expand the discussion of the limitations associated with using a static crop mask in the revised manuscript and include a reference to the recently published MIRCA-OS crop dataset (Kebede et al., 2025). Additionally, we plan to integrate a dynamic crop mask, with a particular focus on the second option discussed above, pending a thorough evaluation of its suitability.*

**Quality Control and Uncertainty analysis:** Government-reported statistics are not always accurate, and the quality of data can vary significantly across countries. Therefore, it is essential to perform quality control and/or assess the uncertainty associated with the data samples—an important step that appears to be missing in the current study.

We agree that yield statistics reported by national governments may vary in quality due to different data collection and aggregation mechanisms as well as reporting accuracies. For a large fraction of the data, we directly rely on reprocessed and curated government-reported statistics. For quality control of these data, we refer to the relevant studies and papers that detail the validation methods applied. Where available, the data card for each country provides a link

to these references. In addition, we apply several quality checks, including filtering out data that fail basic validation criteria, such as non-zero filtering ($yield > 0$) and consistency checks (e.g., ensuring $yield = production/area$). To make clear the limitations of our work, we will add an appendix detailing the QA/QC mechanisms we have in place.

We also acknowledge that there is limited consensus in the community on how to implement quality control and uncertainty analysis for yield data. As highlighted by recent works (Davis et al., 2025), this is an ongoing challenge. A contribution of our work is making these issues visible, and we will continue to update our dataset as better methods and data sources become available.

Nevertheless, we recognize the value of including a quality indicator in the dataset and are committed to incorporating this in a next iteration of our dataset. Possible approaches include assigning a quality tag based on the data source or based on cropping area, which can be obtained from either reported statistics or crop masks.

*In the revised manuscript we will add a discussion on the quality control and uncertainty associated with the yield statistics.*

1. "Crop yield forecasts are produced by both private entities and government institutes using field surveys, process-based crop models, statistical regression and machine learning (Basso and Liu, 2019; Schauberger et al., 2020; Paudel et al., 2021; Gavasso- Rita et al., 2023)": It would be helpful if the authors clearly indicated which citations are associated with each specific method. Also, it is not clear what is the difference between statistical regression and machine learning.

To clarify the citations associated with each method: Basso and Liu (2019) and Schauberger et al. (2020) are review papers that provide an overview of all the mentioned methods. Paudel et al. (2021) focuses specifically on machine learning techniques for crop yield prediction, while Gavasso-Rita et al. (2023) primarily discusses the use of process-based crop models.

Regarding the distinction between statistical regression and machine learning, we will revise the manuscript to provide a clearer explanation. Specifically, statistical regression refers to traditional modeling techniques, such as linear regression, that rely on predefined assumptions about the data (e.g., linearity, normality of errors). While these methods can also be considered part of machine learning, we use the term "machine learning" here to refer to more flexible, data-driven approaches (e.g., random forests, neural networks) that can model complex, non-linear relationships without strong assumptions about the underlying data distribution. We will ensure this distinction is clearly communicated in the revised manuscript.

*We will revise the manuscript as discussed*

2. Line 35: It is biased to only give the example of EU while ignoring other major crop production regions (China, US, South America)

We agree that it is important to provide a more balanced representation of crop yield forecasting practices across different regions. In the revised manuscript, we will expand the discussion to include examples from other major crop production regions, such as China, the US, and South America. This will provide a more comprehensive overview of the global landscape of crop yield forecasting and the diversity of approaches used in different regions.

*We will revise the manuscript as discussed*

3. Line 59: "Benchmark datasets must cover a wide variety of regions and countries": The rationale behind this requirement is unclear. Why is it essential for benchmark datasets to span such a broad geographic scope? I think benchmark datasets are task specific. From the classic computer vision dataset ImageNet to the remote sensing datasets (the UC-Merced dataset (Yang and Newsam, 2010), the WHU-RS19 dataset (Xia et al., 2010), the AID dataset (Xia et al., 2017)), they focus on specific classes/regions/tasks. For the same reason, I think a dataset focusing on U.S. only or China only can still be regarded as benchmarks.

We recognize that several benchmark datasets in computer vision and remote sensing have historically focused on specific tasks, regions, or domains. Our original statement is not to suggest that benchmarks that region-specific datasets cannot be benchmarks, but rather to argue that geographic diversity can be a desirable and valuable property. For agricultural machine learning tasks like crop yield prediction, geographic diversity introduces key scientific and practical challenges. Crop yield is influence by highly variable and regio-specific facors such as climate variability, farming practices, soil properties, and socioeconomic conditions. Datasets limited to a single country or region risk producing models that may not generalize well when applied elsewhere.

Our goal with CY-Bench is to encourage the development and assessment of models that can handle this geographic and contextual variability. Including a wide variety of countries particularly from data-scarce and underrepresented regions that have received little attention in the literature not only improves the robustness and transferability of forecasting models but also promotes research equity. Such efforts allows the community to better understand the specific challenges of forecasting yields in diverse agro-ecological contexts, ultimately advancing global food security research beyond the well-studied regions.

*We will revise our manuscript to clarify this rationale.*

4. Line 110: "we engaged a diverse community of researchers to weigh the benefits and limitations of data sources for each type of data necessary to produce crop yield forecasts" Could the authors be more specific on the decision and quality control processes? What specific benefits and limitations have been considered?

The selection of data sources was guided by several core principles: global coverage, public accessibility, regular updates (except for inherently static datasets), near real-time availability, and their demonstrated relevance to crop growth and development.

In the manuscript, we provide a summary of key decisions, such as the reasoning behind preferring one dataset over another. For a more comprehensive overview, including the specific benefits and limitations of each dataset considered, we have made detailed documentation available on our GitHub repository at `https://github.com/WUR-AI/AgML-CY-Bench/blob/main/data_preparation/DATA-SOURCES-SELECTION.md`. This includes information on the quality control steps taken, as well as a breakdown of the data sources and what each alternative dataset provides. Further, each selected dataset is documented with data cards which contains further links to data sources, related reports and publications.

*We will revise the manuscript to ensure this point is conveyed clearly*

**5. Line 115: "The most relevant weather variables for crop yield forecasting are temperature, solar radiation, and precipitation" It is controversial. Some studies have claimed that VPD and ET are more informative than temperature and precipitation in terms of yield prediction.**

We agree that the relative importance of specific weather variables for crop yield forecasting can vary depending on the study context, crop type, and modeling approach. While temperature, solar radiation, and precipitation are widely recognized as primary drivers, vapor pressure deficit (VPD) and evapotranspiration (ET) are indeed often highlighted as highly informative as well.

That said, VPD and ET are not fully independent variables but are typically derived from the core weather inputs temperature, relative humidity, solar radiation, and wind speed.

In response to the reviewer's suggestion, we will extend the weather predictor set to explicitly include relative humidity and wind speed, enabling users to directly compute VPD and ET if desired.

*We will extend the predictor set to include relative humidity and wind speed, enabling users to compute VPD and ET.*

**6. Line 118: Why was AgERA5 selected? I think there are a lot of alternative choices (PRISM, Gridmet, TerraClimate . . . ) that have been used in yield prediction studies.**

We selected AgERA5 because it offers global coverage (unlike PRISM and GridMET, which cover only the US), daily resolution (unlike TerraClimate, which is monthly), and is specifically tailored to agricultural applications.

A more detailed comparison of candidate datasets and their respective trade-offs is available on our GitHub repository at `https://github.com/WUR-AI/AgML-CY-Bench/blob/main/data_preparation/DATA-SOURCES-SELECTION.md`, and we will include a summary of this evaluation in the revised manuscript.

*We will include our rationale behind selecting AgERA5 in the revised manuscript.*

**7. Line 127: I think SSM and RSM are highly correlated.**

The interaction between these variables is complex and dynamic, influenced by various factors such as soil type, vegetation, and climate. While strong correlation is often observed, divergence can occur (Li et al., 2024). This divergence is particularly interesting as it may reflect changes in soil water dynamics relevant to crop growth and potentially indicate crop stress.

8. Line 145: It is not clear why only fPAR and NDVI are chosen. NDVI has been known for its issue of saturation and there are quite a few alternatives (EVI, GCVI).

We acknowledge that NDVI can saturate in very dense vegetation and that alternatives have been proposed to address this limitation. Our choice to focus on fPAR and NDVI was motivated by their extensive usage in crop yield studies, and their direct connection to canopy structure and light interception, which are key drivers of crop growth. These indicators also provide complementary insights into the underlying dynamics of crop development. While EVI was designed to improve sensitivity in high-biomass regions, it includes a set of empirical constants which requires calibration based on canopy background adjustment to account for the influence of soil and non-vegetated surfaces thus introducing potential inconsistency. That said, we agree that indices like EVI — which reduces saturation effects — and GCVI — which reflects chlorophyll content — can provide complementary insights. Therefore we are open to including them in CY-Bench.

*We will add a discussion on complementary vegetation indices to the revised manuscript.*

9. Line 149: Why using the eight-day composite?

We use the eight-day composite NDVI product to mitigate the effects of cloud cover, which can significantly reduce the reliability of optical remote sensing data. The eight-day compositing process selects the best-quality observation within each period, helping to reduce noise and gaps caused by cloud contamination. This interval offers a well-balanced compromise between temporal resolution and data quality: shorter composites, such as four-day products, tend to be noisier and more affected by clouds, while longer intervals, such as 16-day composites, risk missing short-term vegetation dynamics and phenological changes. For these reasons, the eight-day composite is widely used in crop monitoring applications and provides a practical trade-off for capturing crop development in a consistent and timely manner.

*We will add a short justification for the use of the eight-day composite to the manuscript.*

10. "Predictor data and yield statistics often differ in spatial and temporal resolution, requiring further processing to align them effectively" How would the mismatch between data sources increase the uncertainty and impact the quality of the predictors?

We perform spatial and temporal alignment of predictor variables and yield statistics to ensure that input features correspond accurately with observed outcomes. Spatial aggregation, while potentially causing some information loss in diverse regions, can also smooth out noise, improving the statistical properties of the data used for modeling. For a more elaborate discussion of how such effects influence crop yield modeling, see, for example, Hoffmann et al. (2016); Paudel et al. (2023).

*We will include a brief discussion on how spatial and temporal alignment affects the quality of the predictors.*

11. Line 237 "CY-Bench currently includes predictor data up to and including 2023." What is the starting year?

The starting year is 2003, as one of the predictor datasets, Global Soil Moisture Data from GLDAS, begins from this year, which serves as the limiting factor for the time series.

*We will revise the manuscript to explicitly mention the starting year for clarity.*

12. Figure 3-4: Since the CY benchmark dataset focuses on the sub national yield statistics, it is misleading to color the whole counties, instead of color the sub-national units with the yield records. For example, there is no corn/wheat in US Alaska.

We will update Figures 3 and 4 in the revised manuscript to more accurately reflect the spatial extent of the sub-national yield records. Specifically, we now have thresholded the mean yield records to display only areas where yield data is available (i.e., where yield is greater than zero). Additionally, an animated gif visualizing the yield maps across years is available on our GitHub page, which may help clarify the spatial distribution of the yield data over time.

[Figure]

[Figure]

*We will update Figures 3 and 4 in the revised manuscript to reflect the actual spatial extent of the sub-national yield records.*

13. "The crop masks and crop calendars included in CY-Bench are static, i.e. they do not reflect yearly changes." It is a very critical issue. The cropland experiences dramatic changes from year to year, even at the subnational level. By using a static crop mask, the predictors can

be totally wrong in early years.

We acknowledge the reviewer's critical point regarding the limitations of using a static crop mask in CY-Bench. As mentioned earlier, the availability of consistent, high-quality, dynamic crop masks at a global scale is limited, a constraint shared by many yield prediction and bench-marking efforts. Nevertheless, we recognize the importance of addressing this limitation.

We are committed to improving CY-Bench by incorporating dynamic crop mask sources. Specifically, as noted earlier, we are considering two options: (1) regional, high-resolution dynamic crop masks, which could improve local accuracy but may break harmonization across regions, and (2) the newly released MIRCA-OS crop calendar dataset, which provides global crop masks at 5-year intervals. We plan to integrate this latter dynamic crop mask, pending a thorough evaluation of its suitability.

*In the revised manuscript, we will further elaborate on the limitations of employing a static crop mask. We plan to incorporate the dynamic crop mask from the MIRCA-OS dataset (Kebede et al., 2025).*

14. On the shared Github Leaderboard, there are quite a few counties in which the ML models achieved poor accuracy (negative R2). Given that, how would the authors justify the input features have been well processed, or the uncertainty in the yield data has been well controlled? Could the authors explain if it is safe to use a benchmark datasets whose benchmark accuracies are that low?

The results referenced on the shared leaderboard were produced using an earlier version of the benchmark. Since then, we have made a series of technical corrections and refinements to the data preprocessing workflows, which have led to notably improved prediction accuracy in many regions. The updated results are now available on our GitHub page.

Nevertheless, some countries still show low R2 scores. This is not unexpected, as prediction accuracy can be affected by factors such as the presence of unobserved yield determinants, inherent uncertainty in yield reporting, and a limited number of available training samples, — all of which can vary considerably between countries.

It is important to note that lower performance in some regions should not necessarily be viewed as a flaw, but rather as an indication that the current set of predictors may be insufficient to fully capture the yield variability in those areas. In contrast, the same predictors appear to perform (reasonably) well elsewhere, suggesting that the relevance and completeness of input features can vary depending on the region. Moreover, it prompts a critical reflection on the adequacy of prevailing modeling approaches. Additionally, these patterns could serve as valuable signals for further refinement of yield statistics, helping to enhance the accuracy and reliability of future datasets.

We would like to emphasize that the modeling component remains a work in progress. Our next step with CY-Bench is to prepare a dedicated analysis paper that systematically compares different modeling approaches and explores their performance across regions and conditions.

*We have shared updated results on GitHub. We will prepare an analysis paper that seeks to provide a systematic comparative study of different modeling approaches and discuss their effectiveness across various regional and conditional contexts.*

**References**

Boryan, C., Yang, Z., Mueller, R., and Craig, M.: Monitoring US agriculture: the US department of agriculture, national agricultural statistics service, cropland data layer program, Geocarto International, 26, 341–358, 2011.

Davis, K. F., Anderson, W., Ehrmann, S., Flach, R., Meyer, C., Proctor, J., Ray, D. K., You, L., Foley, M., Kerdiles, H., et al.: HarvestStat: A global effort towards open and standardized sub-national agricultural data, Environmental Research Letters, 2025.

Hoffmann, H., Zhao, G., Asseng, S., Bindi, M., Biernath, C., Constantin, J., Coucheney, E., Dechow, R., Doro, L., Eckersten, H., et al.: Impact of spatial soil and climate input data aggregation on regional yield simulations, PloS one, 11, e0151 782, 2016.

Kebede, E. A., Oluoch, K. O., Siebert, S., Mehta, P., Hartman, S., Jägermeyr, J., Ray, D., Ali, T., Brauman, K. A., Deng, Q., et al.: A global open-source dataset of monthly irrigated and rainfed cropped areas (MIRCA-OS) for the 21st century, Scientific Data, 12, 208, 2025.

Li, N., Skaggs, T. H., Ellegaard, P., Bernal, A., and Scudiero, E.: Relationships among soil moisture at various depths under diverse climate, land cover and soil texture, Science of The Total Environment, 947, 174 583, 2024.

Paudel, D., Marcos, D., de Wit, A., Boogaard, H., and Athanasiadis, I. N.: A weakly supervised framework for high-resolution crop yield forecasts, Environmental Research Letters, 18, 094 062, 2023.

Sweet, L. B., Athanasiadis, I. N., van Bree, R., Castellano, A., Martre, P., Paudel, D., Ruane, A. C., and Zscheischler, J.: Transdisciplinary coordination is essential for advancing agricultural modeling with machine learning, One Earth, https://doi.org/https://doi.org/10.1016/j.oneear.2025.101233, 2025.

Yeh, C., Meng, C., Wang, S., Driscoll, A., Rozi, E., Liu, P., Lee, J., Burke, M., Lobell, D., and Ermon, S.: SustainBench: Benchmarks for Monitoring the Sustainable Development Goals with Machine Learning, in: Thirty-fifth Conference on Neural Information Processing Systems, Datasets and Benchmarks Track (Round 2), `https://openreview.net/forum?id=5HR3vCylqD`, 2021.

---

## Author Comment (AC3)

**Response to Reviewers**

Article Ref.: essd-2025-83

**CY-Bench: A comprehensive benchmark dataset for sub-national crop yield forecasting**

Earth System Science Data

Dear Editor and Reviewers,

We thank you for the insightful suggestions, feedback, and the time you have dedicated to review our manuscript. This is our initial response to your comments. We have provided a point-by-point response below, addressing each specific comment and suggestion. Our actionable items are highlighted in *italics*.

We appreciate your consideration of the effort we made. Please contact us if there are other further suggestions.

Thank you for your kind consideration.

Sincerely,

Authors

**Reviewer #2 comments and suggestions**:

This study proposes CY-Bench, a global, sub-national benchmark for in-season, pre-harvest crop-yield forecasting of maize (covering 38 countries) and wheat (covering 29 countries). The data are collected and curated from large-scale open-access sources. The data are potentially useful for developing and evaluating machine learning models for crop yield forecasting and other Earth system related tasks. In general, CY-Bench is novel and useful for the machine learning and agricultural communities. The following comments should be considered before formally publishing it:

We appreciate your recognition of the novelty and potential value of CY-Bench for both the machine learning and agricultural research communities. We carefully considered your comments and a detailed, point-by-point response to each suggestion is provided below.

1. As CY-Bench is proposed for developing and evaluating data driven models, you should discuss and compare the performance of the models you have benchmarked in the main text, despite a table of model performance is provided in the code repo. You should also discuss

whether there are different findings (of benchmarking these models) compared to existing datasets.

The primary objective of the current study is to introduce and document the CY-Bench dataset, including its scope, geographic coverage, and potential applications. While the performances of a variety of models are provided in the code repository for reference, a comprehensive analysis and discussion of model benchmarking will be addressed in a forthcoming follow-up study. This separation allows us to focus this manuscript on dataset creation, accessibility, and usability, which we believe are the critical contributions of CY-Bench.

A full evaluation of model performance is therefore beyond the scope of this paper. Nevertheless, as detailed in our introduction, in the literature, machine learning methods often outperform classical statistical baselines and this trend does not fully emerge in our experiments. Direct comparisons remain challenging, however, due to non-standardized datasets and evaluation protocols.

*We will revise the manuscript by adding a short note on the initial benchmarking experiments included in the repository. This will briefly acknowledge the availability of baseline results while clarifying that a full and systematic evaluation of model performance will be presented in a separate follow-up study.*

2. Your benchmarking results (as shown in the code repo) displays large variations across different regions. For example, Maize (CN) achieves low NRMSE (8.78) and close-to-one R2 (0.81), while Maize (DK) reaches much higher NRMSE and all very negative R2. This problem can be caused by the models you train or the dataset itself (e.g., quality of the data, predictors you choose), and should be sufficiently discussed in the paper.

The variation in benchmarking results across regions is not unexpected and reflects several underlying factors. In the case of China, yields exhibit a relatively stable spatial pattern, where differences between regions are fairly consistent over time. This makes it relatively easy to achieve high accuracy, especially with location-specific models. For Denmark, the lower performance is likely related to the significantly smaller size of the available dataset, which reduces the predictive signal for the models.

We agree that variability in model performance may also reflect the relevance and completeness of the input features and/or quality of the labels. The fact that the same predictors perform reasonably well in one region while underperforming in others strongly suggests that the explanatory power of these features can vary significantly by location. In our view this finding is not a flaw; it warrants a revisit on the conventional modeling and data selection approaches and offers valuable guidance for assessing the quality of the yield statistics.

We would like to stress that the modeling component and feature importance analysis will be covered in our follow-up analysis paper that systematically compares different modeling approaches and explores their performance across regions and conditions.

*We are preparing a separate analysis paper to systematically explore and compare different*

*modeling approaches, including an examination of the completeness and suitability of input features, building directly on the insights revealed by CY-Bench. Nevertheless, in response to the reviewer's suggestion we will add brief comments on our initial modeling results. In particular, we will note that the benchmarking experiments suggest that the predictive value of the available features differs across regions, leading to variation in model performance. This highlights the complexity of yield prediction across diverse contexts and underscores the importance of CY-Bench as a standardized benchmark to study these differences systematically.*

3. Following the previous comment, I recommend adding a note for each region (or a group of regions) to guide potential users on the specific precautions needed when working with its data (e.g. quality concerns, noise or any other risks).

We agree that region-specific guidance can help users better utilize and interpret the dataset. Our current version already addresses some aspects of this guidance: Figures 5 and 6 show the number of observations per country and the number of administrative regions, highlighting data availability across regions. We also address quality issues with yield labels sourced from government-reported statistics. In the discussion, we highlight risks such as the lack of a temporal crop mask—which may impact feature quality in regions where crop rotation is common—and the lack of separation between irrigated and non-irrigated yields, which could otherwise provide additional insights for data-driven models.

To meet the reviewer's suggestion, we will include notes and provide discussions summarizing key considerations for the use of CY-Bench. These notes will address three key aspects:

First, data quantity: The number of observations varies across regions, which can affect the performance of data-driven models.

Second, input predictor relevance: While our selected predictors generally capture major determinants of yield, their explanatory power arguably varies by region. In some areas, other factors—such as management practices or resource constraints—may more strongly influence yields, which can limit predictive accuracy.

Third, label quality: With respect to yield data, we rely on reprocessed and curated government-reported statistics and apply quality checks to filter implausible values and ensure internal consistency. We note that the community has limited consensus on how to implement quality control and uncertainty analysis for yield data (Davis et al., 2025). Possible approaches include assigning a quality tag based on the data source or on cropping area.

*In the next iteration of the dataset, we plan to provide region-specific notes addressing these aspects wherever relevant. In the revised manuscript, we will also expand the discussion on quality control and uncertainty associated with the yield statistics.*

4. The size of data in each region should be indicated.

The dataset size for each country is presented in Figures 5 and 6, which report both the number of regions and the number of yield observations. These figures also lists the average administrative size of a region within each country.

5. Lines 275 - 280, you should discuss why LOYO is better than random sampling, especially if random sampling was previously more commonly used as you mentioned. As you have noted in limitations, LOYO is a compromise for small datasets; is there a better way for regions with more data?

We used leave-one-year-out (LOYO) cross-validation rather than random sampling to address spatial correlations within the same year. Yields from neighboring regions in the same year are correlated, so random sampling can include similar data in both training and testing sets, violating the independent and identically distributed (IID) assumption and leading to overly optimistic performance estimates.

LOYO tests on an entire year excluded from training, avoiding this issue. It also preserves extreme years in evaluation: the impact of unusually low or high yields is fully represented, rather than being diluted or blended across random splits.

While LOYO is a practical compromise for smaller datasets—allowing maximum use of available training data while ensuring each year is evaluated—regions with larger datasets could benefit from forward sliding (rolling-window) validation, which better mimics operational forecasting. Overall, LOYO provides a balance between maintaining spatial independence, including extreme years in the evaluation, and efficiently using the available data.

*We will elaborate on our choice of LOYO cross-validation compared to random sampling in the revised manuscript, highlighting how it addresses spatial correlations within the same year and preserves the impact of extreme yield years without blending them across splits*

6. In Section 2, how the data are harmonized should be described with more details, instead of simply mentioning it (e.g., Lines 180 - 185 "as they follow a harmonization procedure developed by ..." and Lines 185 - 190 "The data was compiled and harmonized to account for ...").

We will clarify in the revised manuscript that yield datasets used in our study are harmonized to ensure spatial and temporal consistency. For the EU, Ronchetti et al. (2024) harmonized data from multiple sources, standardizing crop definitions, administrative boundaries, and reporting practices, producing comparable annual yield time series. For African countries, the FEWS NET HarvestStat Africa dataset, as compiled and harmonized by Lee et al. (2025), adjusts for changes in administrative boundaries and reporting inconsistencies over time. These harmonization procedures ensure that the annual yield time series are suitable for trend analysis and model evaluation.

*We will elaborate further on the harmonization procedures described above in the revised manuscript*

7. Following the previous comment, you may consider describing more details on how you collect and curate data from a large number of different sources, as this is a major methodological contribution of your study and will be useful for other researchers.

For each data source, both targets (yield), predictors (weather, vegetation, soil) and auxiliary

data (crop mask and crop calendar), we provide detailed data cards in our GitHub repository. These data cards describe provenance, collection, processing, and curation in a transparent and reusable way. Our workflow, which shows how the yield and predictor data is processed, is outlined in Section 2.2 and graphically summarized in Figure 1.

*In the revised manuscript we will expand Section 2.2 to provide a clearer description of how the multi-source data are harmonized, including the steps for temporal alignment to crop seasons and the aggregation to administrative units.*

**References**

Davis, K. F., Anderson, W., Ehrmann, S., Flach, R., Meyer, C., Proctor, J., Ray, D. K., You, L., Foley, M., Kerdiles, H., et al.: HarvestStat: A global effort towards open and standardized sub-national agricultural data, Environmental Research Letters, 2025.

Lee, D., Anderson, W., Chen, X., Davenport, F., Shukla, S., Sahajpal, R., Budde, M., Rowland, J., Verdin, J., You, L., et al.: HarvestStat Africa–harmonized subnational crop statistics for sub-Saharan Africa, Scientific Data, 12, 690, 2025.

Ronchetti, G., Nisini Scacchiafichi, L., Seguini, L., Cerrani, I., and van der Velde, M.: Harmonized European Union subnational crop statistics can reveal climate impacts and crop cultivation shifts, Earth System Science Data, 16, 1623–1649, https://doi.org/10.5194/essd-16-1623-2024, 2024.